# Uncertainty Modeling in Graph Neural Networks via Stochastic Differential Equations

**Richard Bergna**[1,*]**, Sergio Calvo-Ordoñez**[2,3]**, Felix L. Opolka**[4]
**Pietro Liò**[4]**, Jose Miguel Hernandez-Lobato**[1]

[1]Department of Engineering, University of Cambridge
[2]Mathematical Institute, University of Oxford
[3]Oxford-Man Institute of Quantitative Finance, University of Oxford
[4]Department of Computer Science and Technology, University of Cambridge

## Abstract

We address the problem of learning uncertainty-aware representations for graph-structured data. While Graph Neural Ordinary Differential Equations (GNODE) are effective in learning node representations, they fail to quantify uncertainty. To address this, we introduce Latent Graph Neural Stochastic Differential Equations (LGNSDE), which enhance GNODE by embedding randomness through Brownian motion to quantify uncertainty. We provide theoretical guarantees for LGNSDE and empirically show better performance in uncertainty quantification.

## 1 Introduction

Before the widespread of neural networks and the boom in modern machine learning, complex systems in various scientific fields were predominantly modelled using differential equations. Stochastic Differential Equations (SDEs) were the standard approach to incorporating randomness. These methods were foundational across disciplines such as physics, finance, and computational biology [Hoops et al., 2016, Quach et al., 2007, Mandelzweig and Tabakin, 2001, Cardelli, 2008, Buckdahn et al., 2011, Cvijovic et al., 2014].

In recent years, Graph Neural Networks (GNNs) have become the standard for graph-structured data due to their ability to capture relationships between nodes. They are widely used in social network analysis, molecular biology, and recommendation systems. However, traditional GNNs cannot reliably quantify uncertainty. Both aleatoric (inherent randomness in the data) and epistemic (model uncertainty due to limited knowledge) are essential for decision-making, risk assessment, and resource allocation, making GNNs less applicable in critical applications.

To address this gap, we propose Latent Graph Neural Stochastic Differential Equations (LGNSDE), a method that perturbs features during both the training and testing phases using Brownian motion noise, allowing for handling noise and aleatoric uncertainty. We also assume a prior SDE latent space and learn a posterior SDE using a GNN. This Bayesian approach to the latent space allows us to quantify epistemic uncertainty. As a result, our model can capture and quantify both epistemic and aleatoric uncertainties. More specifically, our contributions are as follows:

- We introduce a novel model class combining SDE robustness with GNN flexibility for handling complex graph-structured data, which quantifies both epistemic and aleatoric uncertainties.

- We provide theoretical guarantees demonstrating our model's ability to provide meaningful uncertainty estimates and its robustness to perturbations in the inputs.

---

*Corresponding author: `rsb63@cam.ac.uk`

Workshop on Bayesian Decision-making and Uncertainty, 38th Conference on Neural Information Processing Systems (NeurIPS 2024).

- We empirically show that Latent GNSDEs demonstrate exceptional performance in uncertainty quantification, outperforming Bayesian GCNs [Hasanzadeh et al., 2020], and GCN ensembles [Lin et al., 2022].

## 2 Methodology

Inspired by Graph Neural ODEs [Poli et al., 2019] and Latent SDEs [Li et al., 2020], we now introduce our model: Latent Graph Neural SDEs $-$ LGNSDEs (Figure 1), which use SDEs to define prior and approximate posterior stochastic trajectories for $\mathbf{H}(t)$ [Xu et al., 2022]. Furthermore, LGNSDEs can be viewed as the continuous representations of existing discrete architectures (A.4).

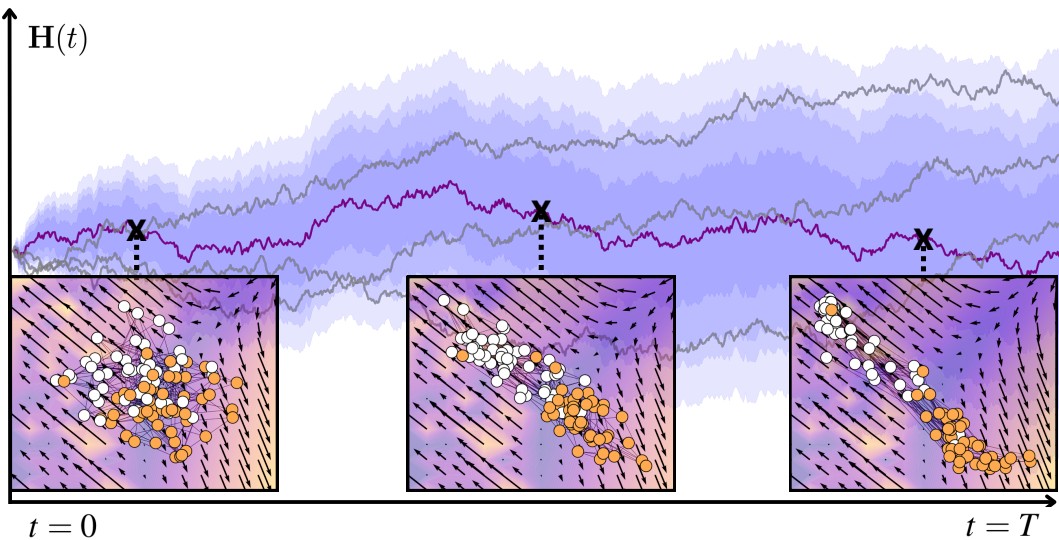

Figure 1: The diagram shows the evolution of one of the nodes of the input graph in latent space, $\mathbf{H}(t)$, through an SDE, with sample paths (purple) and confidence bands representing variance. At three timesteps, we visualize graph embeddings, where nodes (white and orange) become more separable over time due to the influence of the vector field. The inset axes represent latent dimensions, while the purple and yellow background highlights the magnitude and direction of the vector field guiding the latent dynamics.

### 2.1 Model Definition

LGNSDEs are designed to capture the stochastic latent evolution of $\mathbf{H}(t)$ on graph-structured data. We use an Ornstein-Uhlenbeck (OU) prior process, represented by

$$\mathrm{d}\mathbf{H}(t) = \mathbf{F}_{\mathcal{G}}(\mathbf{H}(t), t)\,\mathrm{d}t + \mathbf{G}_{\mathcal{G}}(\mathbf{H}(t), t)\,\mathrm{d}\mathbf{W}(t),$$

where we set the drift and diffusion functions, $\mathbf{F}_{\mathcal{G}}$ and $\mathbf{G}_{\mathcal{G}}$, to constants and consider them hyperparameters. Moreover, $d\mathbf{W}(t)$ is a Wiener process. The approximate posterior is defined as

$$\mathrm{d}\mathbf{H}(t) = \mathbf{F}_{\mathcal{G}}(\mathbf{H}(t), t, \phi)\,\mathrm{d}t + \mathbf{G}_{\mathcal{G}}(\mathbf{H}(t), t)\,\mathrm{d}\mathbf{W}(t), \tag{1}$$

where $\mathbf{F}_{\mathcal{G}}$ is parameterized by a GCN with $\phi$ representing the learned weights of the neural network. The drift function mainly determines the dynamics of the evolution of the latent state, while the diffusion term $\mathbf{G}_{\mathcal{G}}(\mathbf{H}(t))\,\mathrm{d}\mathbf{W}(t)$ introduces stochastic elements. With the need to keep the Kullback-Leibler (KL) divergence bounded, we set the diffusion functions of the prior and posterior to be the same [Calvo-Ordonez et al. 2024, Archambeau et al. 2007].

Let $\mathbf{Y}$ be a collection of target variables, e.g., class labels, for some of the graph nodes. Given $\mathbf{Y}$ we train our model with variational inference, with the ELBO computed as

$$\mathcal{L}_{\mathrm{ELBO}}(\phi) = \mathbb{E}\left[\log p(\mathbf{Y}|\mathbf{H}(t_1)) - \int_{t_0}^{t_1} \frac{1}{2}\|v(\mathbf{H}(u), \phi, \theta, \mathcal{G})\|_2^2\,\mathrm{d}u\right],$$

where the expectation is approximated over trajectories of $\mathbf{H}(t)$ sampled from the approximate posterior SDE, and $v = \mathbf{G}_{\mathcal{G}}(\mathbf{H}(t))^{-1}[\mathbf{F}_{\mathcal{G},\phi}(\mathbf{H}(u), u) - \mathbf{F}_{\mathcal{G},\theta}(\mathbf{H}(u), u)]$.

To sample $\mathbf{H}(t)$ from the approximate posterior, we integrate the SDE in Eq. 1:

$$\mathbf{H}(t_1) = \mathbf{H}(t_0) + \int_{t_0}^{t_1} \mathbf{F}_{\mathcal{G},\phi}(\mathbf{H}(u), u)\, \mathrm{d}u + \int_{t_0}^{t_1} \mathbf{G}_{\mathcal{G}}(\mathbf{H}(u), u)\, \mathrm{d}\mathbf{W}(u),$$

where $\mathbf{H}(t_0)$ are the node-wise features $\mathbf{X}_{\mathrm{in}}$ in the graph $\mathcal{G}$. In practice, this is not feasible since the posterior drift $\mathbf{F}_{\mathcal{G},\phi}$ is parametrised by a neural network. We numerically solve this integral with a standard Stochastic Runge-Kutta method [Rößler, 2010]. We then use a Monte Carlo approximation to get the expectation of $\mathbf{H}(t_0)$ and approximate the posterior predictive distribution as

$$p(\mathbf{Y}^*|\mathcal{G}, \mathbf{X}_{\mathrm{in}}, \mathbf{Y}) \approx \frac{1}{N} \sum_{n=1}^{N} p\left(\mathbf{Y}^*|\mathbf{H}_n(t_1), \mathcal{G}\right),$$

where $\mathbf{H}_1(t_1), \ldots, \mathbf{H}_N(t_1)$ are samples drawn from the approximate posterior $p(\mathbf{H}(t_1)|\mathbf{Y}, \mathbf{X}_{\mathrm{in}}, \mathcal{G})$.

Following Poli et al. [2019], we use a similar encoder-decoder setup. Our encoding focuses solely on the features of individual nodes, while the graph structure remains unchanged. Finally, we remark that the memory and time complexity are $\mathcal{O}(|\mathcal{E}|d + L)$ and $\mathcal{O}(L)$ respectively, where $L$ is the number of SDE solver steps, $\mathcal{E}$ is the number of edges in the graph and $d$ is the dimension of the features.

## 3  Theoretical Guarantees

In this section, we present key results on the stability and robustness of our framework under mild assumptions (Appendix A.2). Firstly, we address the fundamental question of whether our proposed models provide meaningful uncertainties. By showing that the variance of the latent representation bounds the model output variance, we highlight the ability of LGNSDEs to capture and quantify inherent uncertainty in the system. The latent representation is the underlying structure from which the model's output is generated, i.e. the uncertainty in the latent space directly influences the uncertainty in predictions. We formalize this in the following lemma:

**Proposition 1.** *Under assumptions 1-3, there exists a unique mild[1] solution to an LGNSDE of the form*

$$d\mathbf{H}(t) = \mathbf{F}_{\mathcal{G}}(\mathbf{H}(t), t, \boldsymbol{\theta})\, dt + \mathbf{G}_{\mathcal{G}}(\mathbf{H}(t), t)\, d\mathbf{W}(t),$$

*whose variance bounds the variance of the model output $\hat{\mathbf{y}}(t)$ as:*

$$Var(\hat{\mathbf{y}}(t)) \leq L_h^2 Var(\mathbf{H}(t)),$$

*where $L_h^2$ is the Lipschitz constant of the readout layer. This ensures that the output variance is bounded by the prior variance of the latent space, providing a controlled measure of uncertainty.*

We now demonstrate the robustness of our framework under small perturbations in the initial conditions. By deriving explicit bounds on the deviation between the perturbed and unperturbed solutions over time, we show that the model's output remains stable.

**Proposition 2.** *Under assumptions 1-3, consider two initial conditions $\mathbf{H}_0$ and $\tilde{\mathbf{H}}_0 = \mathbf{H}_0 + \delta\mathbf{H}(0)$, where $\delta\mathbf{H}(0) \in \mathbb{R}^{n \times d}$ is a small perturbation in the initial node features with $\|\delta\mathbf{H}(0)\|_F = \epsilon$. Assume that $\mathbf{H}_0$ is taken from a compact set $\mathcal{H} \subseteq \mathbb{R}^{n \times d}$. Then, the deviation between the solutions $\mathbf{H}(t)$ and $\tilde{\mathbf{H}}(t)$ of the LGNSDE with these initial conditions remains bounded across time $t$[2], specifically*

$$\mathbb{E}[\|\mathbf{H}(t) - \tilde{\mathbf{H}}(t)\|_F] \leq \epsilon e^{(L_f + \frac{1}{2}L_g^2)t}.$$

In summary, we show analytically that our framework effectively quantifies uncertainty and maintains robustness under small perturbations of the input. First, we confirm that the model's output variance is controlled and directly linked to the variance of the latent state. Second, we provide a bound on the deviation between solutions with perturbed initial conditions, ensuring stability over time. The proofs can be found in Appendix A.

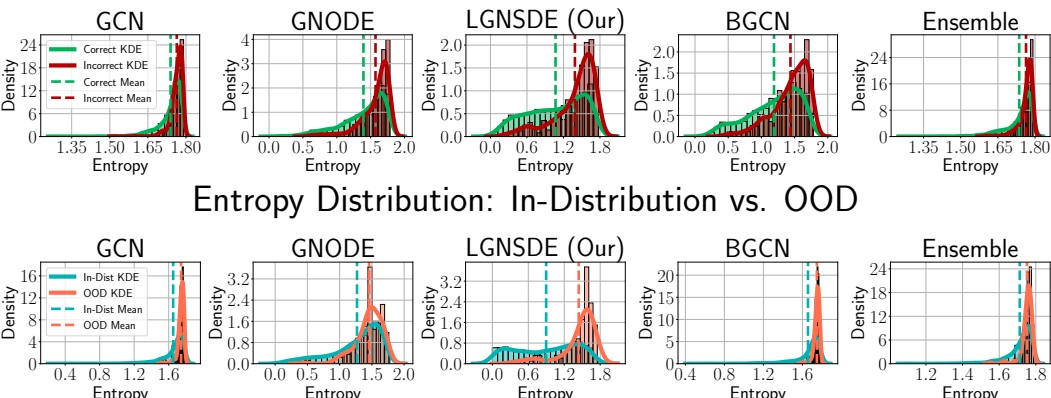

Figure 2: **Top:** Entropy distributions comparing correct and incorrect model predictions on the CORA dataset. Higher entropy is expected for incorrect predictions. **Bottom:** Entropy distributions comparing OOD samples with in-distribution samples in the CORA dataset.

## 4 Experiments

We evaluate LGNSDEs on 5 datasets (see A.2 for details on these datasets and hyperparameters), we compare it to GNODE [Poli et al., 2019], GCN Kipf and Welling [2016], Bayesian GCN (BGCN) [Hasanzadeh et al., 2020], and an ensemble of GCNs [Lin et al., 2022].

The results in Table 3 demonstrate that LGNSDE consistently ranks as either the best or second-best model across most datasets in terms of Micro-AUROC (Area Under the Receiver Operating Characteristic), AURC (Area Under the Risk Coverage), and accuracy. This indicates that LGNSDE effectively handles model uncertainty, successfully distinguishing between classes (AUROC), maintaining low risk while ensuring confident predictions (AURC), and delivering high accuracy.

The top figure 2 shows the entropy distributions of the models for correct and incorrect predictions. Note that most models display similar mean entropy for both correct and incorrect predictions. Notably, our model stands out with the largest difference in entropy, with incorrect predictions having 35% more entropy compared to correct predictions, a larger gap than observed in other models.

### 4.1 Out of Distribution Detection

| Metric | Model | Cora | Citeseer | Computers | Photo | Pubmed |
|---|---|---|---|---|---|---|
| | GCN | 0.7063 ± 0.0569 | 0.7937 ± 0.0366 | 0.7796 ± 0.0271 | 0.8578 ± 0.0136 | 0.6127 ± 0.0351 |
| | GNODE | 0.7398 ± 0.0677 | 0.7828 ± 0.0465 | 0.7753 ± 0.0795 | 0.8473 ± 0.0158 | 0.5813 ± 0.0242 |
| **AUROC (↑)** | BGCN | 0.7193 ± 0.0947 | 0.8287 ± 0.0377 | 0.7914 ± 0.1234 | 0.7910 ± 0.0464 | 0.5310 ± 0.0472 |
| | ENSEMBLE | 0.7031 ± 0.0696 | 0.8190 ± 0.0375 | 0.8292 ± 0.0338 | 0.8352 ± 0.0059 | 0.6130 ± 0.0311 |
| | **LGNSDE (Our)** | 0.7614 ± 0.0804 | 0.8258 ± 0.0418 | 0.7994 ± 0.0238 | 0.8707 ± 0.0099 | 0.6204 ± 0.0162 |
| | GCN | 0.0220 ± 0.0049 | 0.0527 ± 0.0075 | 0.0072 ± 0.0013 | 0.0076 ± 0.0006 | 0.3227 ± 0.0266 |
| | GNODE | 0.0184 ± 0.0053 | 0.0545 ± 0.0110 | 0.0070 ± 0.0029 | 0.0097 ± 0.0015 | 0.3357 ± 0.0309 |
| **AURC (↓)** | BGCN | 0.0208 ± 0.0091 | 0.0458 ± 0.0071 | 0.0064 ± 0.0047 | 0.0108 ± 0.0034 | 0.3714 ± 0.0317 |
| | ENSEMBLE | 0.0215 ± 0.0061 | 0.0487 ± 0.0072 | 0.0041 ± 0.0011 | 0.0081 ± 0.0003 | 0.3277 ± 0.0265 |
| | **LGNSDE (Our)** | 0.0168 ± 0.0070 | 0.0479 ± 0.0109 | 0.0061 ± 0.0011 | 0.0068 ± 0.0008 | 0.3205 ± 0.0135 |

Table 1: AUROC (Mean ± Std) and AURC (Mean ± Std) for OOD Detection across datasets. Red denotes the best-performing model, and blue denotes the second-best-performing model.

We evaluate the models' ability to detect out-of-distribution (OOD) data by training them with one class left out of the dataset. This introduces an additional class in the validation and test sets that the models have not encountered during training. The goal is to determine if the models can identify this class as OOD. We analyze the entropy, $H(y|\mathbf{X}_i) = -\sum_{c=1}^{C} p(y = c|\mathbf{X}_i) \log p(y = c|\mathbf{X}_i)$, where $p(y = c|\mathbf{X}_i)$ represents the probability of input $\mathbf{X}_i$ belonging to class $c$. Entropy quantifies the uncertainty in the model's predicted probability distribution over $C$ classes for a given input $\mathbf{X}_i$.

---

[1]A mild solution to an SDE is expressed via an integral equation involving the semigroup generated by the linear operator and represents a weaker notion of the solution.

[2]Note that while the bound is exponential in $t$, in practice, the time horizon is usually constrained to a limited range, such as $t \in [0, 1]$. Within this interval, the exponential factor does not grow excessively. Furthermore, in practice, $L_g^2 = 0$ and $L_f$ is easily controllable.

Figure 2 shows the test entropy distribution for in-distribution (blue) and out-of-distribution (red) data. For each test sample, predictions were made over $C - 1$ classes, excluding the left-out class. The OOD class exhibits higher entropy, indicating greater uncertainty. While most models show similar entropy distributions for both data types, our LGNSDE model achieves a clear separation, with a 50% higher mean entropy for OOD data compared to in-distribution data. Other models show less than a 10% difference between the two distributions.

Table 1 presents the AUROC and AURC scores for OOD detection across multiple datasets. AUROC evaluates the model's ability to differentiate between in-distribution and out-of-distribution (OOD) samples, with higher scores indicating better discrimination. AURC measures the risk of misclassification as coverage increases, where lower values are preferred. The LGNSDE model (ours) consistently achieves the best AUROC and AURC scores across most datasets, indicating its superior performance in accurately identifying OOD samples and minimizing the risk of misclassification.

## 5 Conclusions and Future Work

In conclusion, LGNSDEs outperform the tested models, opening a new avenue for uncertainty quantification in graph data. In future work, stronger benchmarks should be included for a more comprehensive evaluation. Additionally, Neural SDEs face challenges with time and memory complexity. Further work should explore more scalable sampling methods to address these limitations.

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

## A  Theoretical Remarks

### A.1  Notation

Let $\mathcal{G} = (\mathcal{V}, \mathcal{E})$ denote a graph with node set $\mathcal{V}$ and edge set $\mathcal{E}$. The node feature matrix at time $t$ is $\mathbf{H}(t) \in \mathbb{R}^{n \times d}$, where $n$ is the number of nodes and $d$ is the feature dimension. The evolution of $\mathbf{H}(t)$ is described by a Graph Neural Stochastic Differential Equation, with drift function $\mathbf{F}_{\mathcal{G}}(\mathbf{H}(t), t, \boldsymbol{\theta})$ and diffusion function $\mathbf{G}_{\mathcal{G}}(\mathbf{H}(t), t)$. Here, $\mathbf{F}_{\mathcal{G}}$ depends on the graph $\mathcal{G}$, the node features $\mathbf{H}(t)$, time $t$, and parameters $\boldsymbol{\theta}$. The diffusion function $\mathbf{G}_{\mathcal{G}}$ depends on $\mathcal{G}$ and $\mathbf{H}(t)$ but not on $\boldsymbol{\theta}$, as in practice, this is usually a constant function. The randomness is introduced through the Brownian motion $\mathbf{W}(t)$.

The constants $L_f$ and $L_g$ are Lipschitz constants for the drift and diffusion functions, respectively, ensuring the existence and uniqueness of the solution to the GNSDE. The linear growth condition is controlled by a constant $K$, preventing unbounded growth in $\mathbf{F}_{\mathcal{G}}$ and $\mathbf{G}_{\mathcal{G}}$. Finally, $\mathrm{Var}(\mathbf{H}(t))$ represents the variance of the node features, capturing the aleatoric uncertainty in the system, which is also reflected in the variance of the model output $\mathbf{y}(t) = \mathbf{h}(\mathbf{H}(t))$.

### A.2  Technical Assumptions

**Assumption 1.** *The drift and diffusion functions $\mathbf{F}_{\mathcal{G}}$ and $\mathbf{G}_{\mathcal{G}}$ satisfy the following Lipschitz conditions:*

$$\|\mathbf{F}_{\mathcal{G}}(\mathbf{H}_1(t), t, \boldsymbol{\theta}) - \mathbf{F}_{\mathcal{G}}(\mathbf{H}_2(t), t, \boldsymbol{\theta})\|_F \leq L_f \|\mathbf{H}_1(t) - \mathbf{H}_2(t)\|_F \tag{2}$$
$$\|\mathbf{G}_{\mathcal{G}}(\mathbf{H}_1(t), t) - \mathbf{G}_{\mathcal{G}}(\mathbf{H}_2(t), t)\|_F \leq L_g \|\mathbf{H}_1(t) - \mathbf{H}_2(t)\|_F \tag{3}$$

*for all $\mathbf{H}_1, \mathbf{H}_2 \in \mathbb{R}^{n \times d}$, $t \in [0, T]$, and some constants $L_f$ and $L_g$.*

**Assumption 2.** *The drift and diffusion functions $\mathbf{F}_{\mathcal{G}}$ and $\mathbf{G}_{\mathcal{G}}$ satisfy a linear growth condition:*

$$\|\mathbf{F}_{\mathcal{G}}(\mathbf{H}(t), t, \boldsymbol{\theta})\|_F^2 + \|\mathbf{G}_{\mathcal{G}}(\mathbf{H}(t), t)\|_F^2 \leq K(1 + \|\mathbf{H}(t)\|_F^2),$$

*for all $\mathbf{H} \in \mathbb{R}^{n \times d}$, $t \in [0, T]$, and some constant $K$.*

**Assumption 3.** *The variance of the initial conditions, $\mathbf{H}(0) = \mathbf{H}_0$, of the dynamical system is bounded: $\mathbb{E}[\|\mathbf{H}_0\|_F^2] < \infty$.*

### A.3  Proofs

**Proposition 1.** *Under assumptions 1-3, there exists a unique mild[3] solution to an LGNSDE of the form*

$$d\mathbf{H}(t) = \mathbf{F}_{\mathcal{G}}(\mathbf{H}(t), t, \boldsymbol{\theta}) \, dt + \mathbf{G}_{\mathcal{G}}(\mathbf{H}(t), t) \, d\mathbf{W}(t),$$

*whose variance bounds the variance of the model output $\hat{\mathbf{y}}(t)$ as:*

$$Var(\hat{\mathbf{y}}(t)) \le L_h^2 Var(\mathbf{H}(t)),$$

*where $L_h^2$ is the Lipschitz constant of the readout layer. This ensures that the output variance is bounded by the prior variance of the latent space, providing a controlled measure of uncertainty.*

*Proof.* Using Theorem 1 in Lin et al. [2024], it follows that the Lipschitz conditions of $\mathbf{F}_{\mathcal{G}}$ and $\mathbf{G}_{\mathcal{G}}$ ensure the existence and uniqueness of a mild solution $\mathbf{H}(t)$ to the GNSDE.

Now, consider the stochastic part of the variance of the solution. By applying the Itô isometry, we can compute the expectation of the Frobenius norm of the stochastic integral:

$$\mathbb{E}\left[\left\|\int_0^t \mathbf{G}_{\mathcal{G}}(\mathbf{H}(u), u) d\mathbf{W}(u)\right\|_F^2\right] = \mathbb{E}\left[\int_0^t \|\mathbf{G}_{\mathcal{G}}(\mathbf{H}(u), u)\|_F^2 du\right].$$

Under the Lipschitz condition on $\mathbf{G}_{\mathcal{G}}$, we can bound the variance of $\mathbf{H}(t)$ as follows:

$$\mathrm{Var}(\mathbf{H}(t)) = \int_0^t \|\mathbf{G}_{\mathcal{G}}(\mathbf{H}(u), u)\|_F^2 du.$$

If $\mathbf{G}_{\mathcal{G}}$ is bounded, i.e., $\|\mathbf{G}_{\mathcal{G}}(\mathbf{H}(u), u)\|_F \le M$ for some constant $M$, then $\mathrm{Var}(\mathbf{H}(t)) \le M^2 t$. This shows that the variance of the latent state $\mathbf{H}(t)$ is bounded and grows linearly with time, capturing the aleatoric uncertainty introduced by the stochastic process.

Finally, assuming that the model output $\mathbf{y}(t)$ is a function of the latent state $\mathbf{H}(t)$, $\mathbf{y}(t) = \mathbf{h}(\mathbf{H}(t))$, where $\mathbf{h} : \mathbb{R}^{n \times d} \to \mathbb{R}^{n \times p}$ is a smooth function, we can apply Itô's Lemma as follows:

$$dy(t) = h'(\mathbf{H}(t)) \left[\mathbf{F}_{\mathcal{G}}(\mathbf{H}(t), t, \boldsymbol{\theta}) \, dt + \mathbf{G}_{\mathcal{G}}(\mathbf{H}(t), t) \, d\mathbf{W}(t)\right] + \frac{1}{2} h''(\mathbf{H}(t)) \mathbf{G}_{\mathcal{G}}(\mathbf{H}(t), t)^2 \, dt.$$

For the variance of $\mathbf{y}(t)$, we focus on the term involving $\mathbf{G}_{\mathcal{G}}(\mathbf{H}(t), t) \, d\mathbf{W}(t)$:

$$\mathrm{Var}(\mathbf{y}(t)) = \int_0^t \mathrm{tr}\left(h'(\mathbf{H}(u))^\top \mathbf{G}_{\mathcal{G}}(\mathbf{H}(u), u) \mathbf{G}_{\mathcal{G}}(\mathbf{H}(u), u)^\top h'(\mathbf{H}(u))\right) du.$$

Using the Cauchy-Schwarz inequality for matrix norms, we can bound this as follows:

$$\mathrm{tr}\left(h'(\mathbf{H}(u))^\top \mathbf{G}_{\mathcal{G}}(\mathbf{H}(u), u) \mathbf{G}_{\mathcal{G}}(\mathbf{H}(u), u)^\top h'(\mathbf{H}(u))\right) \le \|h'(\mathbf{H}(u))\|_F^2 \|\mathbf{G}_{\mathcal{G}}(\mathbf{H}(u), u)\|_F^2.$$

Therefore, if $\mathbf{h}$ is Lipschitz continuous with constant $L_h$, then:

$$\mathrm{Var}(\mathbf{y}(t)) \le L_h^2 \int_0^t \|\mathbf{G}_{\mathcal{G}}(\mathbf{H}(u), u)\|_F^2 du = L_h^2 \mathrm{Var}(\mathbf{H}(t)).$$

Hence, under the Lipschitz continuity and boundedness assumptions for the drift and diffusion functions, the solution to the GNSDE exists and is unique, and its output variance serves as a meaningful measure of aleatoric uncertainty. $\square$

**Proposition 2.** *Under assumptions 1-3, consider two initial conditions $\mathbf{H}_0$ and $\tilde{\mathbf{H}}_0 = \mathbf{H}_0 + \delta\mathbf{H}(0)$, where $\delta\mathbf{H}(0) \in \mathbb{R}^{n \times d}$ is a small perturbation in the initial node features with $\|\delta\mathbf{H}(0)\|_F = \epsilon$. Assume that $\mathbf{H}_0$ is taken from a compact set $\mathcal{H} \subseteq \mathbb{R}^{n \times d}$. Then, the deviation between the solutions $\mathbf{H}(t)$ and $\tilde{\mathbf{H}}(t)$ of the LGNSDE with these initial conditions remains bounded across time $t$[4], specifically*

$$\mathbb{E}[\|\mathbf{H}(t) - \tilde{\mathbf{H}}(t)\|_F] \le \epsilon e^{(L_f + \frac{1}{2} L_g^2) t}.$$

---

[3]A mild solution to an SDE is expressed via an integral equation involving the semigroup generated by the linear operator and represents a weaker notion of the solution.

[4]Note that while the bound is exponential in $t$, in practice, the time horizon is usually constrained to a limited range, such as $t \in [0, 1]$. Within this interval, the exponential factor does not grow excessively. Furthermore, in practice, $L_g^2 = 0$ and $L_f$ is easily controllable.

*Proof.* Consider two solutions $\mathbf{H}_1(t)$ and $\mathbf{H}_2(t)$ of the GNSDE with different initial conditions. Define the initial perturbation as $\delta\mathbf{H}(0)$ where $\mathbf{H}_1(0) = \mathbf{H}_0 + \delta\mathbf{H}(0)$ and $\mathbf{H}_2(0) = \mathbf{H}_0$, with $\|\delta\mathbf{H}(0)\|_F = \epsilon$.

The difference between the two solutions at any time $t$ is given by $\delta\mathbf{H}(t) = \mathbf{H}_1(t) - \mathbf{H}_2(t)$. The dynamics of $\delta\mathbf{H}(t)$ are:

$$d(\delta\mathbf{H}(t)) = [\mathbf{F}_\mathcal{G}(\mathbf{H}_1(t), t, \boldsymbol{\theta}) - \mathbf{F}_\mathcal{G}(\mathbf{H}_2(t), t, \boldsymbol{\theta})]\, dt + [\mathbf{G}_\mathcal{G}(\mathbf{H}_1(t), t) - \mathbf{G}_\mathcal{G}(\mathbf{H}_2(t), t)]\, d\mathbf{W}(t).$$

Applying Itô's Lemma to $\text{tr}(\delta\mathbf{H}(t)^\top \delta\mathbf{H}(t))$, we obtain:

$$\begin{aligned}
d(\text{tr}(\delta\mathbf{H}(t)^\top \delta\mathbf{H}(t))) &= 2\text{tr}\left(\delta\mathbf{H}(t)^\top [\mathbf{F}_\mathcal{G}(\mathbf{H}_1(t), t, \boldsymbol{\theta}) - \mathbf{F}_\mathcal{G}(\mathbf{H}_2(t), t, \boldsymbol{\theta})]\right) dt \\
&\quad + 2\text{tr}\left(\delta\mathbf{H}(t)^\top [\mathbf{G}_\mathcal{G}(\mathbf{H}_1(t), t) - \mathbf{G}_\mathcal{G}(\mathbf{H}_2(t), t)]\, d\mathbf{W}(t)\right) \\
&\quad + \text{tr}\left([\mathbf{G}_\mathcal{G}(\mathbf{H}_1(t), t) - \mathbf{G}_\mathcal{G}(\mathbf{H}_2(t), t)]^\top [\mathbf{G}_\mathcal{G}(\mathbf{H}_1(t), t) - \mathbf{G}_\mathcal{G}(\mathbf{H}_2(t), t)]\right) dt.
\end{aligned}$$

Taking the expected value, the stochastic integral term involving $d\mathbf{W}(t)$ has an expectation of zero due to the properties of the Brownian motion. Thus, we have:

$$\begin{aligned}
\mathbb{E}[d(\text{tr}(\delta\mathbf{H}(t)^\top \delta\mathbf{H}(t)))] &= \mathbb{E}\left[2\text{tr}(\delta\mathbf{H}(t)^\top [\mathbf{F}_\mathcal{G}(\mathbf{H}_1(t), t, \boldsymbol{\theta}) - \mathbf{F}_\mathcal{G}(\mathbf{H}_2(t), t, \boldsymbol{\theta})])\right]\, dt \\
&\quad + \mathbb{E}[\|\mathbf{G}_\mathcal{G}(\mathbf{H}_1(t), t) - \mathbf{G}_\mathcal{G}(\mathbf{H}_2(t), t)\|_F^2]\, dt.
\end{aligned}$$

Here, the second term arises from the variance of the diffusion term, as captured by Itô's Lemma. Using the Lipschitz bounds for $\mathbf{F}_\mathcal{G}$ and $\mathbf{G}_\mathcal{G}$, we obtain:

$$\mathbb{E}[d(\text{tr}(\delta\mathbf{H}(t)^\top \delta\mathbf{H}(t)))] \leq \left(2L_f \mathbb{E}[\text{tr}(\delta\mathbf{H}(t)^\top \delta\mathbf{H}(t))] + L_g^2 \mathbb{E}[\text{tr}(\delta\mathbf{H}(t)^\top \delta\mathbf{H}(t))]\right) dt.$$

Rewriting this as a differential inequality:

$$\frac{d}{dt}\mathbb{E}[\text{tr}(\delta\mathbf{H}(t)^\top \delta\mathbf{H}(t))] \leq (2L_f + L_g^2)\mathbb{E}[\text{tr}(\delta\mathbf{H}(t)^\top \delta\mathbf{H}(t))].$$

Solving this using Gronwall's inequality gives:

$$\mathbb{E}[\text{tr}(\delta\mathbf{H}(t)^\top \delta\mathbf{H}(t))] \leq \text{tr}(\delta\mathbf{H}(0)^\top \delta\mathbf{H}(0))e^{(2L_f + L_g^2)t}.$$

Since $\|\delta\mathbf{H}(0)\|_F = \epsilon$, we conclude that:

$$\mathbb{E}[\|\delta\mathbf{H}(t)\|_F] \leq \epsilon e^{(L_f + \frac{1}{2}L_g^2)t}.^5$$

Hence, the deviation in the output remains bounded under small perturbations to the initial conditions, providing robustness guarantees. $\qquad\square$

### A.4 GNSDE as a Continuous Representation of Graph ResNet with Stochastic Noise Insertion

Consider a Graph Neural Stochastic Differential Equation (GNSDE) represented as:

$$d\mathbf{H}(t) = \mathbf{F}_\mathcal{G}(\mathbf{H}(t), t)dt + \mathbf{G}_\mathcal{G}(\mathbf{H}(t), t)d\mathbf{W}(t),$$

where $\mathbf{H}(t) \in \mathbb{R}^{n \times d}$, $\mathbf{F}_\mathcal{G}(\mathbf{H}(t), t)$, and $\mathbf{G}_\mathcal{G}(\mathbf{H}(t), t)$ are matrix-valued functions, and $\mathbf{W}(t)$ is a Brownian motion. The numerical Euler-Maruyama discretization of this GNSDE can be expressed as

$$\frac{\mathbf{H}(t_{j+1}) - \mathbf{H}(t_j)}{\Delta t} \approx \mathbf{F}_\mathcal{G}(\mathbf{H}(t_j), t_j) + \frac{\mathbf{G}_\mathcal{G}(\mathbf{H}(t_j), t_j)\Delta\mathbf{W}_j}{\Delta t},$$

---

[5]Note that the second term (stochastic part) can be omitted as the first term dominates.

which simplifies to

$$\mathbf{H}_{j+1} = \mathbf{H}_j + \mathbf{F}_{\mathcal{G}}(\mathbf{H}_j, t_j)\Delta t + \mathbf{G}_{\mathcal{G}}(\mathbf{H}_j, t_j)\Delta \mathbf{W}_j.$$

Here, $\Delta t$ represents a fixed time step and $\Delta \mathbf{W}_j$ is a Brownian increment, normally distributed with mean zero and variance $\Delta t$. This numerical discretization is analogous to a Graph Residual Network (Graph ResNet) with a specific structure, where Brownian noise is injected at each residual layer. Therefore, the Graph Neural SDE can be interpreted as a deep Graph ResNet where the depth corresponds to the number of discretization steps of the SDE solver.

# B   Details of The Experimental Setup

Table 2: Uniform Hyperparameters of the LGNSDE, GNODE, Bayesian GCN, GCN, Ensemble of GCN models for all datasets.

| Parameter | GNSDE | GNODE | Other |
|---|---|---|---|
| Datasets | All | All | All |
| $t_1$ | 1 | 1 | n/a |
| Hidden Dimensions | 64 | 64 | 64 |
| Learning Rate | 0.01 | 0.01 | 0.01 |
| Optimizer | adam | adam | adam |
| Method | SRK | RK4 | n/a |
| Dropout | 0.2 | 0.2 | 0.2 |
| Diffusion g | 1.0 | n/a | n/a |

| Metric | Model | Cora | Citeseer | Computers | Photo | Pubmed |
|---|---|---|---|---|---|---|
| **MICRO-AUROC (↑)** | GCN | 0.9654 ± 0.0050 | 0.9173 ± 0.0068 | 0.9680 ± 0.0016 | 0.9905 ± 0.0003 | 0.9006 ± 0.0139 |
| | GNODE | nan ± nan | 0.9146 ± 0.0063 | 0.9569 ± 0.0067 | 0.9885 ± 0.0007 | 0.8857 ± 0.0203 |
| | BGCN | 0.9571 ± 0.0092 | 0.9099 ± 0.0090 | 0.9421 ± 0.0097 | 0.9489 ± 0.0189 | 0.7030 ± 0.1331 |
| | ENSEMBLE | 0.9635 ± 0.0031 | 0.9181 ± 0.0062 | 0.9669 ± 0.0025 | 0.9886 ± 0.0004 | 0.8785 ± 0.0163 |
| | **LGNSDE (Our)** | 0.9667 ± 0.0036 | 0.9111 ± 0.0072 | 0.9691 ± 0.0032 | 0.9909 ± 0.0004 | 0.9007 ± 0.0091 |
| **AURC (↓)** | GCN | 0.9966 ± 0.0007 | 0.9966 ± 0.0011 | 0.9994 ± 0.0005 | 0.9987 ± 0.0015 | 0.9994 ± 0.0004 |
| | GNODE | nan ± nan | 0.9967 ± 0.0011 | 0.9994 ± 0.0004 | 0.9998 ± 0.0001 | 0.9915 ± 0.0163 |
| | BGCN | 0.9972 ± 0.0004 | 0.9963 ± 0.0010 | 0.9994 ± 0.0002 | 0.9989 ± 0.0005 | 0.9996 ± 0.0004 |
| | ENSEMBLE | 0.9970 ± 0.0012 | 0.9967 ± 0.0012 | 0.9994 ± 0.0002 | 0.9989 ± 0.0006 | 0.9996 ± 0.0005 |
| | **LGNSDE (Our)** | 0.9970 ± 0.0003 | 0.9971 ± 0.0005 | 0.9995 ± 0.0003 | 0.9997 ± 0.0002 | 0.9995 ± 0.0005 |
| **Accuracy (↑)** | GCN | 0.8105 ± 0.0173 | 0.7258 ± 0.0137 | 0.8098 ± 0.0048 | 0.9116 ± 0.0021 | 0.7570 ± 0.0229 |
| | GNODE | nan ± nan | 0.7235 ± 0.0159 | 0.7911 ± 0.0098 | 0.9053 ± 0.0032 | 0.7577 ± 0.0231 |
| | BGCN | 0.7897 ± 0.0261 | 0.7013 ± 0.0196 | 0.7114 ± 0.0333 | 0.7124 ± 0.0968 | 0.4581 ± 0.1846 |
| | ENSEMBLE | 0.8038 ± 0.0105 | 0.7108 ± 0.0166 | 0.8070 ± 0.0055 | 0.9070 ± 0.0019 | 0.7299 ± 0.0218 |
| | **LGNSDE (Our)** | 0.8113 ± 0.0128 | 0.7120 ± 0.0119 | 0.8247 ± 0.0103 | 0.9169 ± 0.0021 | 0.7595 ± 0.0168 |

Table 3: AUROC (Mean ± Std), AURC (Mean ± Std), and Accuracy (Mean ± Std) for all datasets. Red denotes the best-performing model, and blue denotes the second-best-performing model.

