# OpenReview forum: "Uncertainty Modeling in Graph Neural Networks via Stochastic Differential Equations"
_NeurIPS.cc/2024/Workshop/BDU — NeurIPS BDU Workshop 2024 Poster_

### Official Review · Reviewer_3AHr · 2024-09-20
**The paper presents a novel approach (Latent Graph Neural Stochastic Differential Equations) to uncertainty quantification in graph neural networks. However, a more justification of why these specific approach is well-suited for GNNs is expected to provide and a few mathematical annotations could be improved to enhance clarity with explanations for readers. Overall the theoretical foundation and empirical results look promising and make this a valuable contribution.**

**Rating:** 6
**Confidence:** 3

**Review:**

# Quality

Pros:
- Theoretical foundation: The authors provide a solid mathematical basis for their Latent Graph Neural Stochastic Differential Equations (LGNSDEs) model, including theoretical guarantees (Lemmas 1 and 2).
- Empirical evaluation: The experiment compares LGNSDEs against several baseline models (GCN, GNODE, BGCN, Ensemble) across multiple datasets.
- Performance: LGNSDEs show superior or comparable performance in terms of AUROC, AURC and accuracy.

Cons:
- Mathematical clarity: Some equations and notations lack clear explanations, which results difficulty of understanding for the readers.
- Justification for GNNs: A more detailed explanation of why the proposed SDE formulations are particularly well-suited for Graph Neural Networks should be provided.

# Clarity

Pros:
- The paper is generally well-structured and follows a logical flow.

Cons:
- Most of equations are not numbered. Only 1 question is indexed.
- The drift and diffusion functions are mentioned as constants, consider hyperparameters first, but these functions are then parametrized later depending on a few inputs.
- Some mathematical symbols are not well explained. For example, in the ELBO equation:
$L_{ELBO}(\phi) = \mathbb{E}\left[\log p(Y|H(t_1)) - \int_{t_0}^{t_1} \frac{1}{2} |v(H(u), \phi, \theta, G)|_2^2 du\right]$
The meaning of $v(H(u), \phi, \theta, G)$ is not clearly defined.
- The memory and time complexity is not explained how both are calculated in the appendix.

# Originality

Pros:
- Integration of SDEs with GNNs for uncertainty quantification is innovative and addresses a gap in the field. The ability to quantify both epistemic and aleatoric uncertainties in graph neural networks is promising.
- Theoretical contributions: The provided theoretical guarantees (Lemmas 1 and 2) are novel to the field of uncertainty in GNNs.

# Significance

Pros:
- Novel integration of SDEs with GNNs for uncertainty quantification
- Ability to capture both epistemic and aleatoric uncertainties
- Strong theoretical foundation with proven guarantees

---

### Official Review · Reviewer_DGJq · 2024-09-23
**The paper introduces Latent Graph Neural Stochastic Differential Equations (LGNSDEs), a new approach for modeling uncertainty in graph-structured data.**

**Rating:** 9
**Confidence:** 3

**Review:**

The work is well explained, This novel method combines the concepts from graph neural networks and stochastic differential equations.

Strengths:-
1) Novelty, supported with enough evidence,
2) The paper demonstrates the model's capability in identifying out-of-distribution samples,

Weakness:-
1) It would be better if you can provide the ablations with different hyperparameters, and how it affects the model performance.
2) No mention of computational efficiency
3) I Would like to see the image fonts a bit bigger.[Figure 2]

---

### Decision · Program_Chairs · 2024-10-09

Accept (Poster)